# Development of Low Hysteresis, Linear Weft-Knitted Strain Sensors for Smart Textile Applications

**DOI:** 10.3390/s22197688

**Published:** 2022-10-10

**Authors:** Beyza Bozali, Sepideh Ghodrat, Linda Plaude, Joris J. F. van Dam, Kaspar M. B. Jansen

**Affiliations:** Faculty of Industrial Design Engineering, Delft University of Technology, Landbergstraat 15, 2628 CE Delft, The Netherlands

**Keywords:** smart textiles, knitted strain sensors, hysteresis-free sensors, conductive yarns

## Abstract

In recent years, knitted strain sensors have been developed that aim to achieve reliable sensing and high wearability, but they are associated with difficulties due to high hysteresis and low gauge factor *(GF)* values. This study investigated the electromechanical performance of the weft-knitted strain sensors with a systematic approach to achieve reliable knitted sensors. For two elastic yarn types, six conductive yarns with different resistivities, the knitting density as well as the number of conductive courses were considered as variables in the study. We focused on the 1 × 1 rib structure and in the sensing areas co-knit the conductive and elastic yarns and observed that positioning the conductive yarns at the inside was crucial for obtaining sensors with low hysteresis values. We show that using this technique and varying the knitting density, linear sensors with a working range up to 40% with low hysteresis can be obtained. In addition, using this technique and varying the knitting density, linear sensors with a working range up to 40% strain, hysteresis values as low as 0.03, and *GFs* varying between 0 and 1.19 can be achieved.

## 1. Introduction

With the recent interest in wearable technology, it has become increasingly important to have textile-based sensors that can be a potential platform for future applications and products. Textile-based sensors are essential components of wearable technology due to their flexibility, breathability and ease of application, and therefore can be used in many different applications, such as monitoring heart rate, pulse and other health signals [1], generating human-machine interface systems [2] and producing textile-based systems for electromagnetic interference shielding (EMI) purposes [3]. Textiles with embedded strain sensors can measure local movements of body parts and can be used to monitor posture and limb locomotion but also e.g., respiration. Related applications include sensor gloves and VR suits as well as elbow and knee revalidation aids and sensory compression garments.

A strain sensor is an electronic device that produces an electrical signal in response to mechanical deformation [4]. Typical strain sensors used for engineering purposes consist of thin metal layers on a polyimide substrate and have a working range which is limited to about 1%. Those types of sensors are therefore inadequate to detect movements of the skin or the clothing layer attached to it which may range up to 40% [5]. Textile-based strain sensors offer a new generation of devices that combine strain sensing functionality with wearability, lightness and high stretchability. They can be comfortably worn and sense a wide range of body strains for a vast number of applications.

Textile-based strain sensors can be produced with standard textile manufacturing methods such as knitting [6], embroidering [7], or stitching [8]. For stitched and embroidered sensors, the response of sensors is determined by the stitch structure and as well as the base fabric characteristics, and therefore the elastic response of the substrate is critical to achieving reliable sensors. Knitted structures are good candidates for designing flexible strain sensors thanks to their high stretch recovery and good stretchability characteristics [9]. The main advantage of knitted sensors is that they can be manufactured at low costs in mass production with existing textile equipment and readily available yarns. In addition, the sensors will be washable, breathable, and stretchable while having a fully textile appearance.

The basic working principle of knitted strain sensors can be explained by loops of conductive yarn embedded in overlapping and adjacent rows. During stretching, the points of contact alter, causing the overall resistance to change. This resistance change is reproducible, and if large enough, it can be used as a sensor signal [10]. The human skin typically deforms up to 40% [11] which thus defines the desired upper limit of body strain sensors [12]. For breathing rate measurements, the strains are in the order of 5%. A useful knitted sensor should thus ideally have a working range up to 40% and a resolution of about 1%. In addition, strain sensors for on-body monitoring should be linear and have negligible hysteresis.

The performance of knitted strain sensors is affected by two parameters, one is the material type and the other is the knitting structure. Considering the material type, the effect of elastic or conductive yarns on sensor performance comes to the fore. The embedded structure in a knitted sensor must be highly elastic as it determines the effects of time lag and hysteresis and this elasticity is provided by elastic yarns containing lycra or elastane [13,14]. A study by Atalay pointed out that the properties of elastomeric yarn had a significant impact on the performance of the sensor [10]. In addition, the use of covered elastic yarn instead of bare elastic yarns has resulted in higher sensitivity [15]. Conductive materials are another important determinant of sensor performance. Metallic yarns such as stainless steel-based conductive yarns showed an unstable performance in many studies which was attributed to shortcuts caused by loose yarn filaments. On the other hand, yarns consisting of stainless steel fibers blended with cotton appeared to show improved performance with better sensitivity than sensors knitted with commercial metalized polyamide multifilament yarn due to more stable contact resistance [16]. In contrast to metallic yarns, silver-based nylon yarns show stable electromechanical performance. Although silver-based nylon yarns are capable of producing high sensitivity sensors, the hysteresis values of the sensors can vary from structure to structure and remains a challenge to optimize for reliable sensors. Liang et al. also remark that a promising way to reduce hysteresis is to use Eeontex in the sensor base structure [14]. The resistivities of the currently available conductive yarns vary widely, and the study of Raji et al. [17] seems to suggest that sensors with conductive yarn with a resistance of 77 ohm/cm have an optimum *GF* in the resistance range of 40 to 120 ohms, but the effect was not studied in detail [18]. The type of knit design is another essential parameter that affects the sensor performance, and many different studies to investigate the effect on sensor performance were reported. One is a single jersey knit strain sensor, developed by adding float and crimp stitches to a structure knit that appears to be suffering from relatively large hysteresis effects [19]. Interlock structures knitted with conductive yarns show a good sensor performance having a gauge factor (*GF*) value of 3.5 at between a 10–40% working range. In the case of 1 × 1 rib knitted samples, the effect of the aspect ratio and shape of the sensing region was evaluated by Raji et al. and they concluded that plain rectangular sensors showed higher repeatability and lower noise levels compared to non-rectangular shaped sensors [15]. A 1 × 1 rib knit design was found to be promising as a strain sensor and presented a sensor with a *GF* of 1.36 and a working range of 0–21% by Chia et al. [20].

Various materials and methods in the design of knitted strain sensors have been reported so far but until now are not reliable, and linear and low-hysteresis sensors were reported. In our quest to develop such sensors we take the findings of previous studies, summarized below as a starting point:Steel-based conductive yarns are less suitable for strain sensors because of the relatively high noise levels associated with knitted sensors and silver coated nylon yarns should be preferred accordingly [16,21].Next, the base fabric in which the sensor yarns are embedded needs to be fully elastic. If not, the sensor will not respond elastically and will be prone to show time lag and hysteresis effects [14].Rib structures as knit design allow for stretching abilities up to 100% and are therefore potential strong candidates for knitted strain sensors [17].Sensors should preferably be rectangular with aspect rations between 24:1 to 77:1 [17].

In the current paper, we take these conclusions as a starting point and report a systematic study on the performance of knitted strain sensors to develop reliable strain sensors with high sensitivity, wide working range and low hysteresis. We do this by manufacturing a series of sensors with the elastic and conductive yarn types as variables, the number of conductive courses, the knit density, and the conductive yarn position in a co-knitted configuration. The electromechanical performance is evaluated by determining the initial resistance, sensitivity, working range and hysteresis for all knitted sensors. Based on these sensor constructions, knitted sensors with minimum hysteresis, maximum sensitivity and working range are selected and evaluated.

## 2. Materials and Methods

Based on the above-mentioned observations from the literature, the following initial choices are made: (i) A rectangular sensor area consisting of a number of parallel conductive courses, (ii) A 1 × 1 rib knit design, (iii) Silver-coated or plated nylon as the conductive yarn, and (iv) Elastic yarn with as high elastic recovery as possible. As elastic yarns, we used the ones provided by Yeoman (Nm 13, 81% Nylon and 19% Lycra) and Uppingham (Nm 11, 16.74% Lycra and 83.26% Nylon). As conductive yarns, six different silver-coated nylon yarns with various resistivities from Amann Threads UK Ltd. (Ashton-under-Lyne, UK) and Shieldex, Statex Productions- and Sales GmbH (Bremen, Germany) were selected (See the optical images of the conductive yarns in Appendix A). The property details and related coding of these yarns are listed in Table 1.

In weft knitted fabrics, the horizontal row of continuously connected loops is called a course, whereas the series of loops that intermeshes vertically is known as wales, as illustrated in Figure 1a. In this study, knitted strain sensors have been produced on a Stoll CMS 530 flatbed knitting machine with a gauge of E8. An 1 × 1 rib knit structure was chosen for all samples. The conductive yarn is co-knitted with a non-conductive elastic yarn using a yarn carrier with two eyelets as shown in Figure 1b. In that way, the position of the conductive yarn with respect to the elastic support yarn is well controlled, resulting in a rib structure in which the conductive yarn is always on the inside or always on the outside. A more detailed explanation is given in Section 3.4. In order to enhance the elasticity, the base fabric was also manufactured with two parallel elastic yarns. All sensor samples have outer dimensions of approximately 14.5 × 7 cm. The size of the conductive area changes according to the number of conductive courses, which varied between 1 to 5.

Knitted strain sensors were prepared with the parameters listed in Table 2. The conductive course number was varied between 1 to 5, and knit densities were varied by selecting NP numbers between 6 to 15. The NP value is related to the needle pitch distance. The position of the conductive yarn is referred to as conductive outside (CO) and conductive inside (CI), respectively. This will be discussed in detail later (see Appendix A).

In this paper, knitted samples will be coded as NP${}_{x}$Y${}_{x}$C${}_{x}$E${}_{x}$ following all the parameters listed in the Table 1 and Table 2. As an example, NP${}_{10}$Y${}_{S1}$C${}_{2}$E${}_{1}$ can be interpreted as the sample knitted with a NP setting of 10 and with two conductive courses using S${}_{1}$ conductive and E${}_{1}$ elastic yarns, respectively.

### 2.1. Resistance-Strain Testing Procedure

The electromechanical properties of the knitted strain sensors were investigated by measuring the changes in resistance during cyclic tensile tests in which the sample was subsequently loaded and unloaded. A custom-made tensile tester with a 1000 N load cell was used to generate a cyclic stretching between 0–40% at a constant speed of 30 mm/min as depicted in Figure 2. The samples were stretched in the course direction. Preliminary testing showed that test speed did not have a large effect on the performance in terms of sensitivity and working range in Appendix A. The initial length of the samples known as the gauge length was set at 50 mm. All tests were conducted in standard atmospheric conditions. The resistances were measured using a 4-point probe method in which a constant current value of 0.3 mA was applied to the outer red clamps in Figure 2. The resulting voltage drop was read from the inner dark green electrodes and stored in a data acquisition system. The resistance values were calculated as the ratio between measured voltage and current values. Mechanical clamps with conductive Velcro tape were used to clamp the samples and provide the electrical connection.

### 2.2. Electromechanical Properties

The electromechanical properties such as gauge factor *GF*, working range *WR* and hysteresis were investigated to analyze the performance of the knitted sensors. During the stretching of sensors, the resistance does not always change monotonically. Thus not all parts of the recorded resistance changes are suitable to indicate the sensor performance. We define the working range as the region in which the resistance changes monotonically. The onset and end of the working range are referred to $\varepsilon_{max}$ and $\varepsilon_{min}$, respectively (see Figure 3a. The working range (*WR*) is then defined as (Equation (Equation 1));
(1)$$Working\;Range\left(WR\right)=\varepsilon_{max}-\varepsilon_{min},$$

The *GF* or sensitivity is defined as the slope of the linear range of the relative resistance change versus the applied strain graph as can be seen in Figure Equation 3a. Equation (Equation 2) shows the formula for the gauge factor in which $\Delta R=R_{Stretched}-R_{0}$, $R_{0}$ is the initial resistance, and Δε is the strain change within the linear region of the curve.
(2)$$Gauge\;Factor\left(GF\right)=\frac{\left(\Delta R/R_{0}\right)}{\Delta \varepsilon },$$

Hysteresis values can be calculated along the resistance axis (vertical) or along the strain axis (horizontal), and these values will be referred to as *resistance hysteresis* and *strain hysteresis*, respectively [18]. For a strain gauge in an application, the expected errors in the measured strain are of primary concern; thus, as a performance indicator, we prefer to use the strain hysteresis values and not the more commonly reported resistance hysteresis [22,23]. When the maximum strain in a loading cycle increases, it is evident that the hysteresis also increases. Since we prefer the hysteresis values not to depend too much on the loading conditions, we choose to scale the hysteresis with the working range as shown in Equation (Equation 3) and Figure 3b. The hysteresis strain, $\Delta \varepsilon_{hysteresis}$, can be defined as the maximum difference between loading and unloading strain as shown in Figure 3b.
(3)$$H_{\varepsilon }=\frac{\Delta \varepsilon_{hysteresis}}{WR}.$$

## 3. Results

### 3.1. Electromechanical Performance Analysis of a Knitted Strain Sensor

As a first example, we discuss the NP${}_{1}{}_{0}$ knitted sample test as shown in Figure 4.

During stretching up to 40%, the relative resistance of the sample changes from 0 to a value of 26%. The last part of the stretching curve shows that the amount of increase in the resistance gradually decreases. During the first part of the unloading curve, the resistance roughly follows the curve of the loading cycle, but deviations start below about a 20% strain, leading to hysteresis. At the end of the unloading cycle, the resistance change approaches nearly its starting value. For this experiment, the working range, gauge factor and hysteresis are determined as *WR* = 0–27%, *GF* = 0.89 ± 0.06, and *H* = 0.07, respectively.

As previously explained in Section 2, knitted sensors were fabricated by making conductive/elastic yarn combination. Knitted sensors without elastic support yarn were also investigated to better understand the effect of using elastic yarn as a co-knit yarn in the sensing region. The results showed that the sample without the co-knitted elastic yarn had a nearly strain-independent resistance where *GF* was close to −0.1, which can be seen in Appendix A. A gauge factor below zero means that the resistance drops when the sample is stretched. Configurations with gauge factors close to zero may prove useful as stretchable interconnects in knitted textile circuits.

### 3.2. The Effect of Knit Density (NP Number) on the Sensor Performance

Knitted densities are determined by the NP numbers. A higher NP number results in less dense fabric structures, while lower NPs result in denser fabrics. Optical images of the NP${}_{x}$Y${}_{S1}$C${}_{2}$E${}_{1}$ series are shown in Figure 5. Areal densities (g/m${}^{2}$) of the knitted fabrics are calculated using Equation (Equation 4) in which YL is the yarn length in a knit design (mm), LD denotes the linear density in Tex, WS is the wale spacing in mm, CS the course spacing in mm, $S_{h}$: horizontal size, and $S_{v}$: vertical size [24]. The yarn length was determined by unraveling the course of the fabric and measuring the distance when stretched.
(4)$$Areal\;Density=\frac{YL\times LD}{WS\times S\times S_{h}\times S_{v}},$$

The optical images show that, when the NP number increases, the samples become looser and less dense, an observation which is conformed by the decrease in areal density in relation to the NP numbers as can be seen in Table 3. Accordingly, the conductive yarns in the sensing area do not fully integrate into the structure when the NP numbers increase from NP${}_{10}$ to NP${}_{15}$, which could play an active role in the sensor performance in terms of the elastic recovery.

The electromechanical performance of the strain sensors knitted with different NPs changing from NP${}_{6}$–NP${}_{15}$ is shown in Figure 6 along with *GFs* and working ranges. Tighter structures knitted with the NP numbers 6, 7, and 8 have a working range between 0–10%; above that strain value, the performance of sensors becomes unstable and nonlinear (see Appendix A). The response curve for the NP${}_{9}$ sensor showed an initial minimum resistance of 15.5 ohms, hence the effective working range did not start at 0 but at a 6% strain. Similar to the working range, the gauge factor values are relatively low and range between 0.20 and 0.33. However, for less dense sensors, the gauge factors improve significantly and range from 0.70 to 1.91 as the NP number changes from 9 to 15. When the NP numbers increased from NP${}_{9}$ to NP${}_{15}$, the working range became broader.

Table 3 shows the areal density and measured elastic modulus (E) of the knitted samples with several NP numbers. Nominal thickness values were calculated by following the standard of BS 2544:1954—determination of thickness of the textile fabric. By using the Heidenhain length gauge, fabric samples were placed between two glass plates and fabric thicknesses were calculated by subtracting the total thickness from the previously measured glass plate thickness. The elastic modulus values were calculated by averaging over three tests. The results show that the areal density values decrease with the increase in the NP number. With an increase in NP number from 6 to 15, the areal density decrease with a factor 5.4, the fabric thickness increases 1.6 times and the modulus decreased with a factor 8.7. It demonstrates that, with higher NP numbers, we obtain more open structures with a lower areal density and a lower elastic modulus and, as seen in Figure 6, an increased working range and gauge factor.

### 3.3. The Effect of an Elastic Yarn Type on Sensor Performance

The hysteresis of knitted strain sensors is known to be directly related to the ability of mechanical recovery of the sample. Therefore, in this section, we investigate the effect of elastic yarn type on sensor performance by separately comparing the performance of sensors with elastic yarns of type E${}_{1}$ and E${}_{2}$. The test results are summarized in Figure 7. Although both yarns still have hysteresis, the samples knitted with E${}_{1}$ yarn appear to be more promising for each NP numbers except NP${}_{8}$. Due to their higher elastic recovery performance and relatively lower hysteresis values in comparison with the E${}_{2}$ yarn, for all next samples, the E${}_{1}$ type yarn was used. In addition, we choose to select the NP${}_{9}$ as the preferred density setting in all subsequent tests. Even though the sensor knitted with NP${}_{7}$ and NP${}_{1}{}_{0}$ show the lowest hysteresis, a larger improvement in hysteresis was achieved for the NP${}_{9}$ sample with the manufacturing technique we used for low-hysteresis sensors and the NP${}_{9}$ sample seems to be more promising. This will be discussed at later stages in more detail.

### 3.4. The Effect of Conductive Yarn Configuration on the Sensor Performance

As explained in Section 2, we knitted the conductive yarns together with an elastic support yarn. By switching the position of elastic and conductive yarns in the yarn carrier, we can control whether the conductive yarn always ends up on top or below the elastic yarn (see Appendix A). For a standard jersey knit, this would result in one fabric surface with mainly conductive yarn and one with a mainly elastic yarn. The 1 × 1 rib structure that we use in this study, however, creates a more complex structure in which, in an unstretched state, the conductive yarns ends up either at the inside or the outside of the fabric. In this section, we will investigate how that positioning affects the sensor performance. Whether the conductive yarn has been positioned inside or outside, the knit structure will be referred to as CO and CI. Examples are shown in Figure 8.

The comparison of NP${}_{9}$Y${}_{S1}$C${}_{2}$E${}_{1}$CO and NP${}_{9}$Y${}_{S1}$C${}_{2}$E${}_{1}$CI samples in terms of electromechanical performance is illustrated in Figure 9. Preliminary experiments showed that, for all knitted samples, the first resistance cycle always deviated significantly which was attributed to an initial rearrangement of the knitted structure. Therefore, in all subsequent graphs, the first cycle will be omitted. In Figure 9, the resistance change graphs of both a sample with the conductive yarns outside (CO, Figure 9a) and one with the yarn inside (CI, Figure 9b) are presented. This clearly shows that the CI sample performs significantly better regarding hysteresis and linearity. The working range changes from 4–32% for the CO sample to 0–40% for the CI variant. In addition, the resistance increases almost linearly over the full working range for the CI sample, whereas the resistance change flattens out above about 28% in the case of the CO sample. Additionally, the *GF* of the CI sample, 1.19 ± 0.04, is higher than that of the CO knitted sensor with a value of 0.70 ± 0.002. The most significant difference was observed in the hysteresis values, which shows a sharp decline from 0.15 to 0.03 of CO and CI, respectively. The graphs show data from four subsequent cycles (cycles 2 to 5), which can be seen to be nicely overlapping indicating that at least during these initial tests the gauges showed negligible drift.

The electromechanical performance of the NP${}_{9}$Y${}_{S1}$C${}_{2}$E${}_{1}$CI and NP${}_{9}$Y${}_{S1}$C${}_{2}$E${}_{1}$CO sensors at high strain values was also investigated and compared by testing the sample up to 70% strain to determine the maximum of the working range. The resistance versus strain graph is depicted in Figure 10. Figure 10a shows that the resistance change levels-off between strain values 40% and 70%, indicating an overall working range of 0–40% when the conductive yarn is inserted inside. Figure 10b shows the nonlinear behaviour of the CO sample with a higher hysteresis when the deformation is applied.

### 3.5. The Effect of Conductive Yarn Type and Conductive Course Number on Sensor Performance

Knitted strain sensors were produced using Shieldex (S) yarns with conductive course numbers ranging from 1 to 5 with a fixed NP${}_{9}$ setting and both conductive inside (CI) and outside (CO) structures. The results of both S${}_{1}$ and S${}_{2}$ samples were illustrated in Figure 11 with gauge factor and hysteresis values. The average and corresponding error bars are presented to indicate the spread in data. For 1-course samples of S${}_{2}$, only a single measurement was available, and consequently no error value was reported. In Figure 11a,c, the gauge factor is plotted versus the number of conductive courses using conductive yarn type S${}_{1}$ and S${}_{2}$, respectively. In all cases, the conductive inside (CI, grey bars) configuration resulted in the highest gauge factor values, with the effect being largest for the S${}_{2}$ yarn (Figure 11c). For both yarn types, 4-course samples appear to be the optimum, resulting in *GF* values of 1.92 ± 0.24 and 2.10 ± 0.22 for S${}_{1}$ and S${}_{2}$, respectively.

The corresponding hysteresis values are plotted in Figure 11b,d. Once again the conductive inside configuration produces the best results with lowest hysteresis values. For S${}_{1}$ yarn, the change from CO to CI results in a drop in hysteresis from about 0.14 on average for the CO samples to about 0.06 for the CI samples. The lowest hysteresis values are close to *H*${}_{\varepsilon }$ = 0.04 and are observed for a 2-course sample (S${}_{1}$ yarn, CI) and 3- or 4- course samples (S${}_{2}$ yarn, CI). Note that these values are scaled to the working range, according to our definition in Equation (Equation 3). As a result, the configuration in which the conductive inside improves the performance not only for the specific knitted sample discussed in Section 3.4, but also for different conductive yarns and course numbers.

By comparing Figure 11b,d, it shows that the hysteresis value is not much affected by the number of conductive courses and always stays at a relatively high value of around 0.16 for CO samples with the S${}_{1}$ yarn and around 0.20 for the S${}_{2}$ yarn.

After evaluating the effect of yarn positioning on sensor performance for the S${}_{1}$ and S${}_{2}$ yarns of Shieldex for both CI and CO samples, the question was to explore if the conductive course numbers have an effect on CI samples knitted with Amann yarns. Samples were prepared and compared with six different conductive yarns with the yarn specifications listed in Table 1. For the comparison, only 1- and 2-conductive course numbers were prepared with CI samples and all samples, were knitted with the NP${}_{9}$ setting. The gauge factor versus initial resistance plots are shown in Figure 12.

The general trend is that the gauge factor decreases with increasing the initial resistance (R${}_{0}$). The major deviation from this trend is for the A${}_{1}$ yarn, which shows a relatively low *GF* at low initial resistances for both 1- and 2- conductive course samples. It should be noted that yarn A${}_{1}$ is much thicker than the support elastic yarn E${}_{1}$, which has a thickness of 0.39 ± 0.02 mm, and this may have affected the stability of the configuration during co-knitting (see Appendix A and Table 4).

Overall, it can be seen that 2-course samples showed better sensitivity in comparison with the 1-course samples. Thinner yarns with high resistivity such as S${}_{2}$ and S${}_{3}$ show a relatively poor sensing performance with *GF* values below 0.4. The highest *GF* observed is that for the 2-course configuration of the S${}_{1}$ yarn around 1.19 and A${}_{2}$ around 0.79. Note that low initial resistances result in higher power consumption, which can be important in practical applications for smart textiles [13].

## 4. Discussion

The electromechanical performance of knitted strain sensors was studied for different knitting densities, elastic/conductive yarn types, course number and elastic/conductive co-knit configurations. The sensing performance of the knitted strain sensors was evaluated with the main focus on gauge factor and hysteresis. As it can be seen from Figure 13, by varying the NP number (knitting density), we have control over the sensitivity while still being able to keep the hysteresis low.

The gauge factors corresponding to the curves for NP${}_{6}$, NP${}_{9}$ and NP${}_{10}$ are 0.20 ± 0.06, 1.19 ± 0.04 and 0.89 ± 0.06, respectively. Note that knitted conductive yarn structures with gauge factors close to zero could be of large practical importance as a basis for stretchable interconnections between sensors and data processing units in a smart garment application. The ability to control the strain sensitivity as shown in Figure 13 thus would enable future researchers to knit a garment with a network of connected sensor elements in one go and on one machine. Note that a stretchable conducting line with zero gauge factor could also be obtained by omitting the elastic support yarn in the co-knit structure (see the black line in Figure 13).

The role of the elastic support yarn below the conductive component is therefore probably to provide enough contact pressure between overlapping loop segments. If this pressure is absent and overlapping loops do not act as shortcuts in the electrical path, the mechanism of sliding contacts and changing path lengths is no longer applicable, and no resistance changes are observed. Considering our interpretation that the sliding of overlapping loops contacts is responsible for obtaining resistance changes, we could interpret the observed knit density effects by realizing that in closer knitted structures the sliding may be partially impaired by adjacent yarns. The larger the NP, the more space for contact point sliding and, consequently, the larger resistance change and gauge factor [25].

The hysteresis values are affected not only by the NP number but also by the position of the conductive yarn. To understand the reasoning behind achieving less hysteresis sensors by positioning conductive yarn inside the structure, we need to understand the loop structure under deformation. Resistance changes due to stretching a knitted structure can in principle be due to both the deformation of the structure and to the stretching of the conductive yarns itself. However, due to the large loops in the structure and the relatively high stiffness of the conductive yarns, the structural deformation will be the dominant part. Only at extremely large deformations (above 70%), the loop lengths are sufficiently straightened up to allow the loading and stretching of the conductive yarns. Note that this type of loading may cause cracks and decrease the adhesion between silver and polyamide in the silver coating, and thus compromises the long-term sensor reliability [26]. The fact that this type of yarn stretching in our 1 × 1 rib structure only occurs at extremely high deformations is therefore highly beneficial for the sensor reliability.

In Figure 14, we depict how the co-knitting of a conductive yarn on top of a non-conductive one can result in a structure in which the conductive yarns end up on the inside. Note that this type of configuration is typical for the 1 × 1 rib structure and is not observed for other knitting stitches. When stretched out, the conductive parts of the interlaced loops make contact and start to slide relative to each other which causes the resistance change responsible for the sensor function [27]. In general, when the fabric is stretched, the intra-yarn contact points are pulled apart resulting in increased resistance, whereas the inter-yarn contact points are pulled together leading to reduced resistance.

Knitted sensors with a variety of conductive yarns show different sensor performances. Since the elastic and conductive yarn is inserted into the structure together, the yarn thickness can be another parameter that affects the performance. All yarn thicknesses were estimated using an image analysis program and the thickness of the elastic yarn E${}_{1}$, as well as the six conductive yarns, are listed in Table 4. From this, we can conclude that the highest sensitivity values can be achieved when the conductive yarns have comparable thickness compared to the elastic yarn. Note that the difference between the Shieldex yarns and Amann yarns is that Shieldex yarns consist of only silver coated nylon filaments, whereas Amann yarns are hybrid filaments with polyester fibers.

In the results presented until now, the best performing sensor was the one with a gauge factor and hysteresis of 1.19 and 0.03, respectively. Our developed sensor can be compared with the other research in literature as shown in Table 5. In addition, these types of sensors are suitable with high signal integrity required applications such as a respiration sensor or finger monitoring applications [28]. Higher gauge factors were observed for the series, as seen in Figure 6 at around 1.91 and for the series with four courses at around 2.10 (see Figure 11). Such that the NP${}_{15}$ sensor with lower resistance and high elasticity can be used for application in which pauses are needed during the middle of motion monitoring such as dance movements [14].

## 5. Conclusions

This paper investigates the effect of structure and process parameters on knitted strain sensor performance by conducting a systematical study. Knitted strain sensors have the potential to be used in various applications for body movement monitoring and therefore they should have specific features in terms of sensitivity and stability.

The findings of this study can be summarized in several points:The separation of the loops within the knitted structure produced with lower NPs such as 6, 7 and 8 cannot be achieved directly because of the tightness of the samples. At higher NP values of 12 and 15, the sensor starts to respond at higher strain values thanks to an improved elastic response for strains up to 40%.A manufacturing technique in which the sensor properties, especially the hysteresis values, can be reduced and controlled in knitted sensors has been reported for the first time. By positioning the conductive yarns inside the 1 × 1 rib structure, a knitted strain sensor with linear performance and a gauge factor of 1.19 and negligible hysteresis of 0.03 was developed.The effect of yarn positioning on different types of conductive yarns in terms of hysteresis was also proved. For both S${}_{1}$ and S${}_{2}$ yarns, the hysteresis values have decreased for different conductive course numbers. The lowest hysteresis values were achieved for a 2-course sample of S${}_{1}$ yarn and 3- or 4- course samples of S${}_{2}$ yarn.The performance of sensors made with different conductive yarns was evaluated, and those with lower initial resistance reported higher gauge factors for S-type yarns.The highest sensitivity values can be achieved when the conductive yarns have comparable thickness compared to the elastic yarn. Thinner yarns result in poor sensing performance.

## Figures and Tables

**Figure 1 sensors-22-07688-f001:**
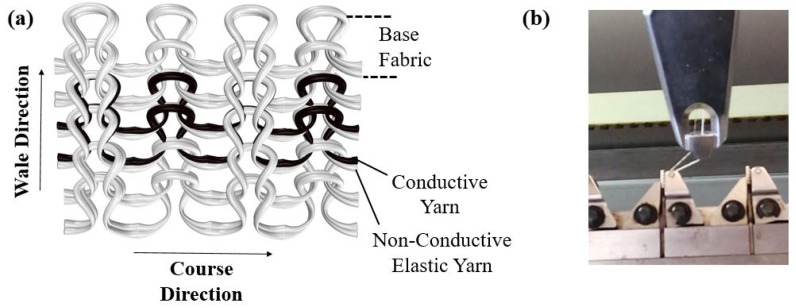
(**a**) Co-knitted 1 × 1 rib structure with two conductive yarn courses: (indicated in black) and non-conductive yarns (white), and (**b**) A yarn carrier containing two parallel yarns.

**Figure 2 sensors-22-07688-f002:**
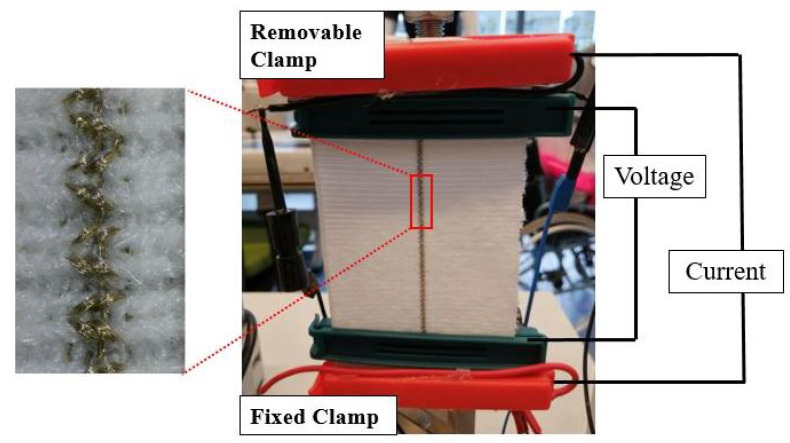
Test setup with an attached knitted strain sensor and four-point probe connections.

**Figure 3 sensors-22-07688-f003:**
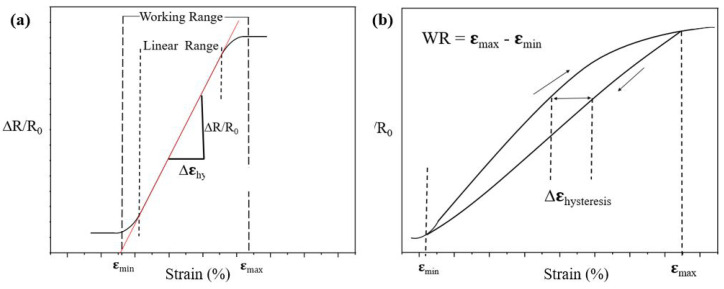
(**a**) A graph showing how to calculate the gauge factor and working range, and (**b**) A graph showing how to calculate the resistance hysteresis (Black arrows represent the loading and unloading during a cycle.

**Figure 4 sensors-22-07688-f004:**
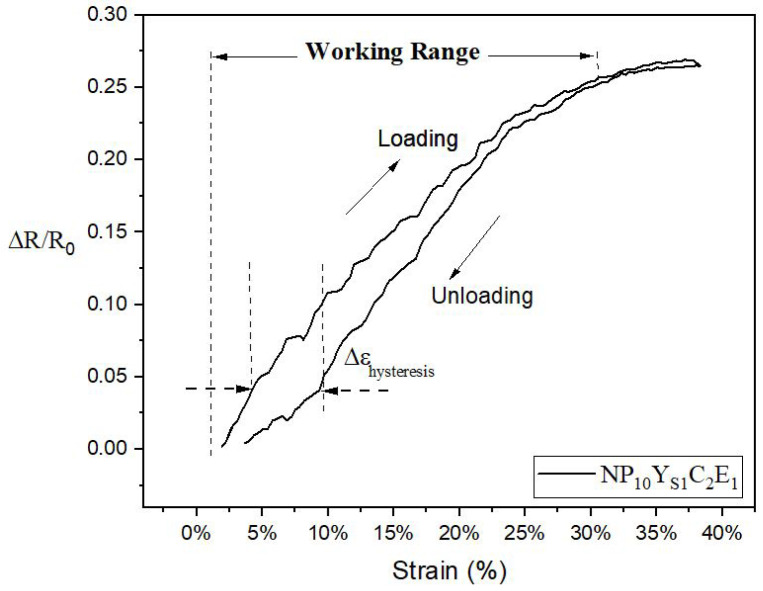
Relative resistance change versus strain plot of knitted strain sensor type NP${}_{10}$Y${}_{S_{1}}$C${}_{2}$E${}_{1}$.

**Figure 5 sensors-22-07688-f005:**
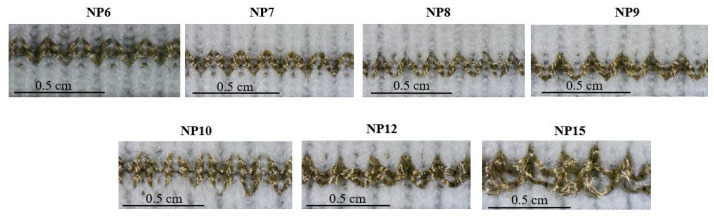
The optical images of knitted strain sensors of the series of NP${}_{x}$Y${}_{S1}$C${}_{2}$E${}_{1}$.

**Figure 6 sensors-22-07688-f006:**
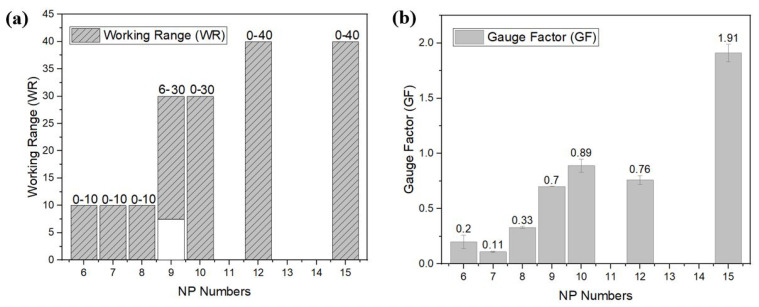
The effect of NP values on the sensor performance for the series of NP${}_{x}$Y${}_{S_{1}}$C${}_{2}$E${}_{1}$ (**a**) working range, and (**b**) gauge factor.

**Figure 7 sensors-22-07688-f007:**
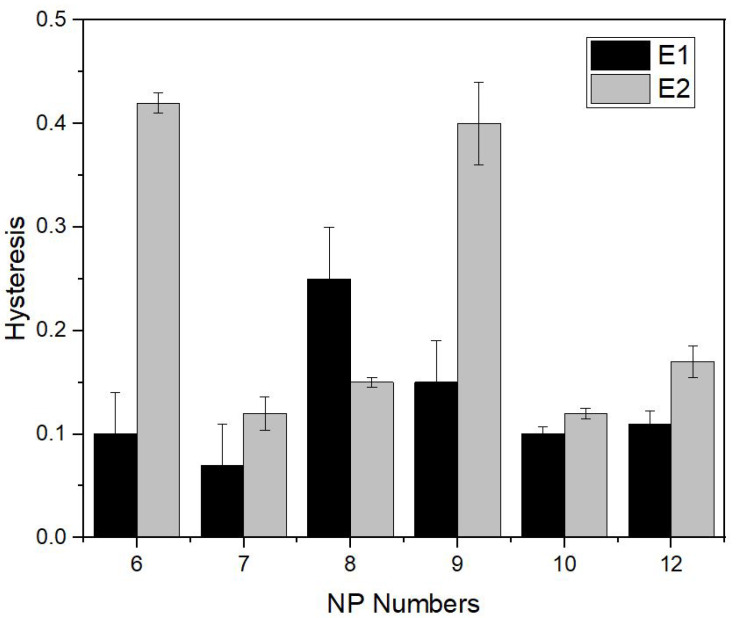
The effect of elastic yarn type (E${}_{1}$ and E${}_{2}$) on hysteresis for the series of NP${}_{x}$Y${}_{S1}$C${}_{2}$E${}_{1}$ and NP${}_{x}$Y${}_{S1}$C${}_{2}$E${}_{2}$. (Hysteresis is calculated according to Equation (Equation 3)).

**Figure 8 sensors-22-07688-f008:**
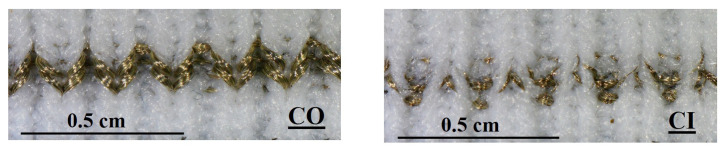
The yarn configuration of two samples (**a**) NP${}_{9}$Y${}_{S1}$C${}_{2}$E${}_{1}$CO and (**b**) NP${}_{9}$Y${}_{S1}$C${}_{2}$E${}_{1}$CI.

**Figure 9 sensors-22-07688-f009:**
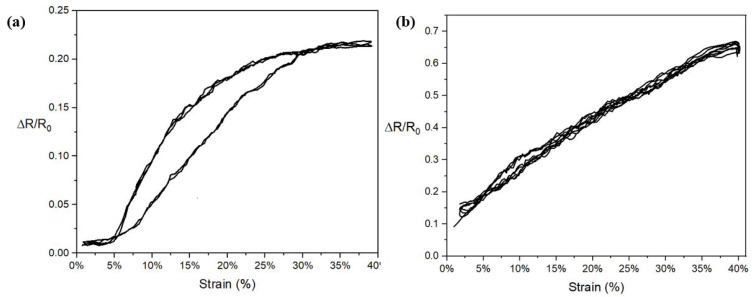
The two different results of samples knitted with conductive outside (**a**) NP${}_{9}$Y${}_{S1}$C${}_{2}$E${}_{1}$CO and conductive inside (**b**) NP${}_{9}$Y${}_{S1}$C${}_{2}$E${}_{1}$CI.

**Figure 10 sensors-22-07688-f010:**
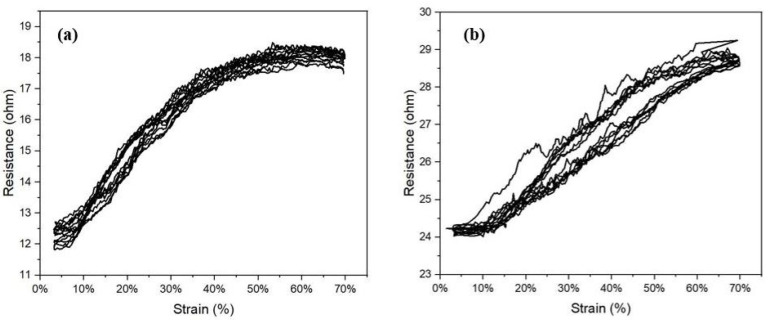
The electromechanical behaviour of the (**a**) NP${}_{9}$Y${}_{S_{1}}$C${}_{2}$E${}_{1}$C*I* and (**b**) NP${}_{9}$Y${}_{S_{1}}$C${}_{2}$E${}_{1}$C*O* sensor sample at a 70% strain value over eight applied cycles.

**Figure 11 sensors-22-07688-f011:**
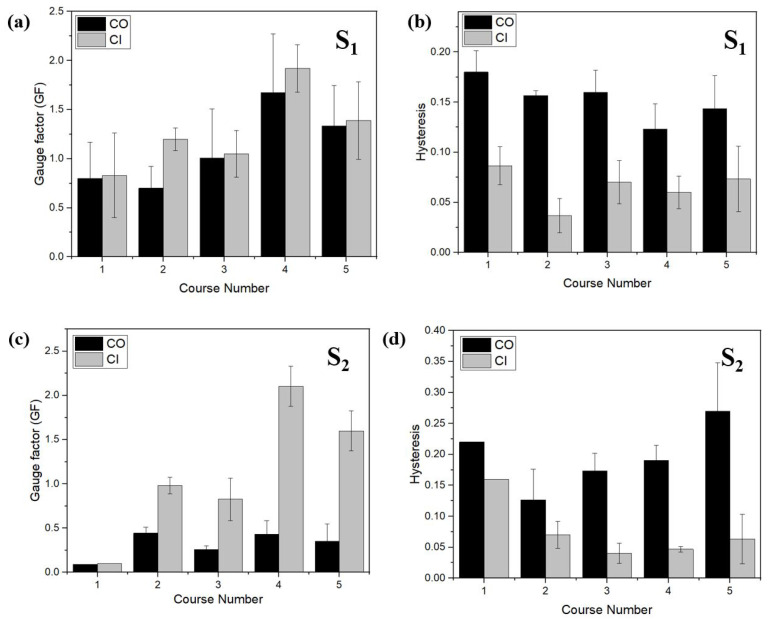
Effect of course number and yarn position S${}_{1}$ yarn and S${}_{2}$ yarn, respectively; (**a**) Gauge factor vs course number, (**b**) Hysteresis vs course number, (**c**) Gauge factor vs course number, and (**d**) Hysteresis vs course number.

**Figure 12 sensors-22-07688-f012:**
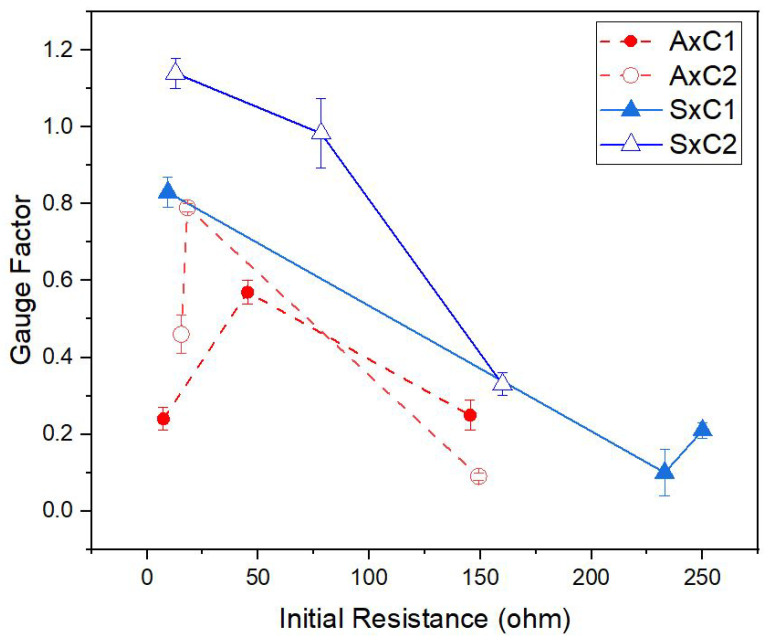
Gauge factor versus initial sensor resistance plots of sensors knitted with conductive yarn inside (CI) and NP${}_{9}$; Sensors knitted with Shieldex yarns (S${}_{1}$, S${}_{2}$ and S${}_{3}$) are indicated with blue straight lines, and Amann yarns (A${}_{1}$, A${}_{2}$ and A${}_{3}$) with red dashed lines.

**Figure 13 sensors-22-07688-f013:**
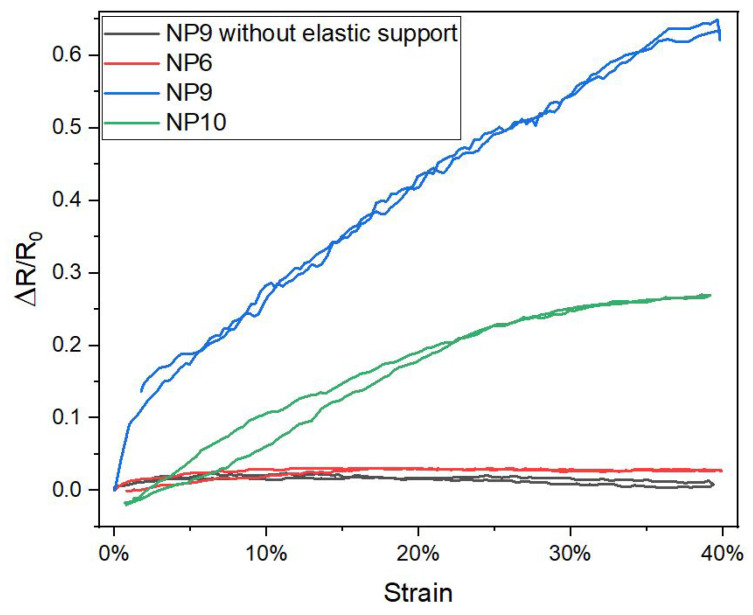
Different graphs showing high and low hysteresis knitted samples; NP${}_{9}Y_{S1}C_{2}$—without E1, with NP${}_{6}$, NP${}_{9}$ and NP${}_{10}$.

**Figure 14 sensors-22-07688-f014:**
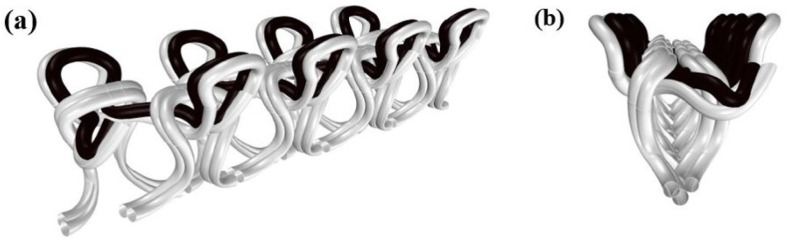
(**a**) The illustration of the knitted sample which conductive yarn inserted inside and elastic yarn positioned outside (CI), (**b**) Representation of the unfolded loops in a course which shows the relative position of the conductive (black line) and elastic (grey line) yarns.

**Table 1 sensors-22-07688-t001:** Conductive yarn details and related properties.

Composition	Coding	Linear Density (Dtex *)	Resistivity (Ω/m)	Manufacturer
Silver coated	A1	960	85	
polyamide/	A2	620	150	Amann
polyester hybrid yarn	A3	280	530	
Silver	S1	235	600	
plated	S2	117	1500	Shieldex
polyamide 6 yarn	S3	78	3500	

* The weight per 10,000 m (dtex) reflects the yarn thickness.

**Table 2 sensors-22-07688-t002:** Parameters of knitted strain sensors tested in the study and the section in which they are discussed.

Parameters	Variations	Section
NP number	6, 7, 8, 9, 10, 12 and 15	Section 3.2
Conductive yarn type	Shieldex (S${}_{1}$, S${}_{2}$, S${}_{3}$) and Amann (A${}_{1}$, A${}_{2}$, A${}_{3}$)	Section 3.5
Conductive course number	1, 2, 3, 4 and 5	Section 3.5
Elastic yarn type	Yeoman (E${}_{1}$) and Uppingham (E${}_{2}$)	Section 3.3
Configuration of the c. yarn	Inside (CI) and Outside (CO)	Section 3.4

**Table 3 sensors-22-07688-t003:** The areal density and elastic modulus of the knitted sample series of NP${}_{x}$Y${}_{S1}$C${}_{2}$E${}_{1}$.

NPs	Areal Density (g/m${}^{2}$)	Nominal Thickness (mm)	Elastic Modulus (MPa)
6	796	1.57 ± 0.02	0.26 ± 0.01
7	614	1.57 ± 0.03	0.24 ± 0.02
8	551	1.76 ± 0.01	0.21 ± 0.005
9	402	1.74 ± 0.01	0.15 ± 0.01
10	331	1.84 ± 0.01	0.12 ± 0.01
12	216	2.27 ± 0.005	0.06 ± 0.005
15	148	2.53 ± 0.005	0.03 ± 0.002

**Table 4 sensors-22-07688-t004:** The yarn thicknesses and electromechanical performance of sensors knitted with NP${}_{9}$ and CI/CO configurations.

Yarn Type	Yarn Thickness (mm)	Gauge Factor CO CI
A1	0.49	0.24 0.46
A2	0.33	0.57 0.79
A3	0.27	0.25 0.09
S1	0.32	0.70 1.19
S2	0.19	0.44 0.98
S3	0.16	0.21 0.33

**Table 5 sensors-22-07688-t005:** The performance comparison of previous literature studies and this study.

Ref.	Knit Type	Conductive Material	Non-Conductive Material	GF	Hysteresis	WR
Atalay (2013) [10]	Interlock	235 dtex Silver- Nylon	800 dtex elastic yarn	3.75	0.10	10–40%
Atalay (2014) [29]	Interlock	Silver-Nylon	800 dtex elastic yarn	3.44	0.08	>40%
Tognetti (2014) [30]	Intersia	Carbon filled nylon	Lycra	6.7	0.14	>10%
Raji (2018) [15]	1 × 1 Rib	Silver-Nylon	Nylon covered spandex	1.25	-	32.8
Raji (2018) [15]	1 × 2 Rib	Silver-Nylon	Nylon covered spandex	1.50	-	25.3
Raji (2018) [15]	1 × 3 Rib	Silver-Nylon	Nylon covered spandex	1.60	-	31.7
Oks (2014) [31]	Plain	Silver-Nylon	Elastane, Pes, Cotton	10	0.10	5–10%
Chia (2021) [20]	1 × 1 Rib	235 dtex Silver-Nylon	Spandex yarn	1.36	-	20%
Yang (2009) [32]	1 × 1 Rib	Bare SS	-	−1.1	-	50%
Our Study (2022)	1 × 1 Rib	235 dtex Sil-Ny, 600 **Ω**/m	Nylon- Lycra	1.19	0.03	40%

## Data Availability

The data presented in this study are available on request from the corresponding author. The data are not publicly available due to patent is pending.

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
