# Peer review of "Development of Low Hysteresis, Linear Weft-Knitted Strain Sensors for Smart Textile Applications"

_sensors, 2022, doi:10.3390/s22197688_

Round 1
Reviewer 1 Report
The comments file is attached below.

Author Response
Dear Reviewer,
We sincerely acknowledge the reviewers for their professional and constructive comments. In response to the reviewers’ comments, we hereby submit the revised manuscript entitled “DEVELOPMENT OF LOW HYSTERESIS, LINEAR WEFT-KNITTED STRAIN SENSORS FOR SMART TEXTILE APPLICATIONS”.
In the following pages the reviewers’ comments are addressed one by one. Particularly, the discussion section has been thoroughly reviewed addressing the comments of the referees (e.g. adding table xx); this is detailed in the response to the reviewers’ comments, cf. below. The responses are written in red, and the revised text that will be added to the manuscript will be written both in red and highlighted in yellow. You can find the responses in the attached pdf file.
We hope that the newly added or modified parts in the article together with the information provided in this letter will meet the expectations and that the revised article can be published in the sensors MDPI journal.
Sincerely,
Beyza Bozali (corresponding author)
TU Delft | Faculty of Industrial Design Engineering
Department of Sustainable Design Engineering

Reviewer 2 Report
The authors proposed the method to evaluate the effect of structure and electromechanical properties on knitted strain sensor performance. The research approach is very innovative and interesting. Nevertheless, the following specific questions should be answered and added to the manuscript to improve its clarity.
There are some comments listed below:
1. Please describe the size of conductive yarn and Elastic yarn with the number of cores. “Course number” in Figure 11 should be “Number of cores.” Please attach pictures of yarn with one core and five cores.
2. In section 2, please arrange the parameters of table 2 in the order defined NPxYxCxEx. Please add the definition of Emax and Emin in Formula 1. In formula 3, please add the calculation of delta hysteresis.
3. In section 3, why is NP8 the exception? How many samples are used for each NPx? The author explained that choosing E1 due to low hysteresis, why was NP9 selected as the preferred density setting for all subsequent tests, while except for NP8, NP9 has the highest hysteresis?
4. Figures 5 and 8 show that the scale bar does not seem in 3cm. Please check!
5. All figures should have an average value with deviation. Please modify!
6. Figure 10 is missing data for CO. Please modify it!
Author Response

(The authors gave the same response as above.)

Reviewer 3 Report
Generally speaking this is a very good paper. It can be (almost) published in its present form. I have two very minor comments:
1) line 151: table S1, should be table 1
2) fig 3a: the subscript "hysteresis" must be deleted.
I have a more general comment but the authors do NOT have to change their paper to take this comment into account. Several hysteresis loops but did you also measure a hysteresis loop in the plot stress versus strain? The area of such a loop is the amount of heat dissipated in the fabric. If the fabric is stretched periodically, you have a periodic heat generation. Now my question: is it possible that the loops you presented (fig.3c...) are due to the dependence of the electric resistance on temperature?
Author Response

(The authors gave the same response as above.)

Round 2
Reviewer 2 Report
I am satisfied with the answers and additional explanations of the authors in the revised version of this manuscript and I suggest it be accepted.